# Pareto Prompt Optimization

**Guang Zhao[1], Byung-Jun Yoon[1,2], Gilchan Park[1]**
**Shantenu Jha[3,4,5], Shinjae Yoo[1], Xiaoning Qian[1,2]**

[1]Brookhaven National Laboratory, [2]Texas A&M University, [3]Princeton Plasma Physics Laboratory
[4]Rutgers University–New Brunswick, [5]Princeton University
`gzhao@bnl.gov, bjyoon@tamu.edu, gpark@bnl.gov,`
`shantenu@pppl.gov, sjyoo@bnl.gov, xqian@ece.tamu.edu`

## Abstract

Natural language prompt optimization, or prompt engineering, has emerged as a powerful technique to unlock the potential of Large Language Models (LLMs) for various tasks. While existing methods primarily focus on maximizing a single task-specific performance metric for LLM outputs, real-world applications often require considering trade-offs between multiple objectives. In this work, we address this limitation by proposing an effective technique for multi-objective prompt optimization for LLMs. Specifically, we propose *ParetoPrompt*, a reinforcement learning (RL) method that leverages dominance relationships between prompts to derive a policy model for prompts optimization using preference-based loss functions. By leveraging multi-objective dominance relationships, Pareto-Prompt enables efficient exploration of the entire Pareto front without the need for a predefined scalarization of multiple objectives. Our experimental results show that ParetoPrompt consistently outperforms existing algorithms that use specific objective values. ParetoPrompt also yields robust performances when the objective metrics differ between training and testing.

## 1 Introduction

Advancements in Large Language Models (LLMs) have attracted significant interest due to their remarkable capabilities across various natural language processing (NLP) tasks. Prompting, a method that utilizes a natural language prefix or context to guide the LLMs to complete a desired task, allowing us to utilize LLMs' capabilities without re-training LLMs (Wei et al., 2022; Wen et al., 2024; Reynolds & McDonell, 2021). However, crafting effective prompts often necessitates significant manual efforts, requiring expertise in both LLMs and the specific task domains (Wang et al., 2024).

Prompt optimization emerged as a powerful solution, leveraging algorithms to automate the search for optimal prompts. These algorithms encompass diverse techniques such as gradient-based optimization (Wen et al., 2024), reinforcement learning (RL) (Deng et al., 2022; Zhang et al., 2022), evolutionary algorithms (Zhou et al., 2022), beam search (Pryzant et al., 2023), and inverse RL (Sun et al., 2023). These methods typically formulate prompt optimization focusing on single objectives, aiming to optimize a single chosen performance metric, such as accuracy or fluency. In real-world applications, however, prompt effectiveness often involves trade-offs between multiple objectives. For example, a prompt designed for text style transform may need to balance style consistency with content accuracy, while a prompt for summarizing factual topics may need to consider both informativeness and conciseness.

Unlike single-objective formulations (Wen et al., 2024) where we may obtain a clear "best" prompt, in multi-objective prompt optimization problems with multiple conflicting objectives, there is not a single prompt that excels in all objectives simultaneously. Instead, we aim for a set of prompts from the so-called *Pareto front*, which represents the best possible trade-offs between these objectives. To search for and maximally cover the Pareto front, optimization algorithms—including RL or evolutionary algorithms often utilize an indicator function or reward function to evaluate the quality of prompts. These solution strategies translate the multi-objective performance into a single value that reflects how good a set of prompts is, for example, via the weighted sum of objectives, S metric or

Hypervolume of the dominated region (Baumann & Kramer, 2024), and product of multiple objectives (Jafari et al., 2024). The introduction of these indicator/reward functions in *ad-hoc* ways helps guide prompting towards the Pareto front. But it also imposes rigid assumptions about the trade-offs between objectives, which oversimplifies the nuanced preferences involved in text generation. For example, the weighted sum of objectives assumes the linear trade-offs between objectives and the weight is predefined. While S-metric assumes a uniform preference across all regions of the objective space. Some objective of text generation can be measured by various metrics, such as fluency by perplexity or grammaticality, each of which may have non-linear relationships. The previously mentioned weighted sum and S-metric fail to capture these complexities, as they overlook the possibility of different preferences arising from evaluating the same objective by varying metrics.

In this study, rather than using a scalar metric to describe the multi-objective performance of prompts, we propose to guide the prompt search by comparing pairs of prompts based on fundamental principles of multi-objective problems. Specifically, if one prompt dominates another, the dominating prompt is considered more **preferable**. Conversely, if a pair of prompts do not dominate each other, then we do not prioritize one over the other. This approach ignores the specific objective values of the prompts. While discarding specific values might seem to reduce the algorithm's effectiveness, two key factors motivate our approach. First, in language generation tasks, accurate and reliable evaluation of absolute objective values is often unavailable or unreliable. Relative preferences, on the other hand, are more accessible and robust in handling the inherent vagueness in evaluation objectives. Second, using dominance relationships avoids imposing assumptions on the underlying structure of the objectives. This also eliminates the need to assume the additive contributions of objectives, uniform preferences across regions, or biases introduced by reference points.

Based on these motivations, we propose ***ParetoPrompt***, a novel multi-objective prompt optimization method driven by preference based RL. In our formulation, prompts are generated by a policy model. During each iteration, the algorithm samples pairs of prompts for the same input instance and compares their dominance relationship, then update the policy model accordingly. The algorithm can be combined with various types of RL-based prompt generation methods, such as direct text token generation from language models (Deng et al., 2022; Wu et al., 2022), or using RL-trained edit agents (Zhang et al., 2022). We have conducted experiments comparing ParetoPrompt with competing baselines. Our results clearly show that, despite solely based on dominance relationships, ParetoPrompt achieves better or comparable performance than the algorithms relying on specific objective values. Additionally, our method demonstrates robust performance even when the training metric differs from the evaluation metrics used during testing. The code for our implementation is made available at https://github.com/guangzhao27/ParetoPrompt.

## 2 RELATED WORK

We first review existing research on Prompt Optimization and Multi-objective Optimization for LLMs, including the relevant recent work on Direct Preference Optimization (DPO).

### 2.1 PROMPT OPTIMIZATION

Prompting has become a prevalent approach for guiding LLMs towards specific tasks within the NLP domain. Soft prompting techniques require access to the latent embeddings, limiting their applicability in closed-source LLMs (Li & Liang, 2021). Natural language prompt optimization leverages optimization algorithms to generate effective text prompts without modifying LLM parameters (Wen et al., 2024; Deng et al., 2022; Zhang et al., 2022; Zhou et al., 2022; Pryzant et al., 2023; Sun et al., 2023). For instance, Lin et al. (2024) learn a reward model trained on human preference data and then optimize the reward model to find the optimal prompt. Fernando et al. (2023); Guo et al. (2023) employ evolutionary algorithms for single-objective prompt optimization. However, a majority of these works focus on single-objective optimization formulations.

### 2.2 MULTI-OBJECTIVE OPTIMIZATION FOR LANGUAGE MODELS

The field of multi-objective optimization has been explored for both prompt optimization and LLM fine-tuning. Baumann & Kramer (2024) proposed an Evolutionary Algorithm (EA) where LLMs perform "crossover" and "mutation" operations on prompts during the optimization process. Jafari

et al. (2024) adapted various scalar reward functions (e.g., HyperVolume Indicator, Expected product of objectives) for multi-objective prompt optimization using RL. Jang et al. (2023) introduced a method for fine-tuning LLMs for multiple objectives by training separate policy models and merging their parameters for personalized preferences. Zhou et al. (2023) proposed Multi-Objective Direct Preference Optimization (MODPO), an RL-free algorithm extending Direct Preference Optimization (DPO) for datasets with multiple dimensional preference. They prioritised preferences with weight vectors and derived a multi-objective preference reward function. Our work distinguishes itself from these previous efforts by utilizing dominance preference signals between prompt pairs, avoiding the need for predefined scalar metrics or reward functions, which often introduce ad-hoc assumptions of the objectives.

## 2.3 DIRECT PREFERENCE OPTIMIZATION

DPO (Rafailov et al., 2024) offers an alternative to RL for fine-tuning pre-trained language models. It utilizes human preference data directly for updates instead of training a reward model like in Reinforcement Learning from Human Feedback (RLHF). Azar et al. (2024) proposed Identity Preference Optimization (IPO), a generalization of DPO, that replaces a non-decreasing function related to the Bradley-Terry model (Bradley & Terry, 1952) with the identity function, which mitigates overfitting issues in DPO. The prompt optimization problem is different from the DPO fine-tuning problem. Prompt optimization depends on the specific task and language model in use, leading to the lack of a widely accepted general prompt dataset for prompt optimization. Therefore, we adopt a RL approach to interact with the task-specific language model, enabling us to learn the most effective prompts for particular problems. Our work utilizes the DPO/IPO reward function for dominance preference data in a multi-objective optimization framework.

## 3 PARETO PROMPT OPTIMIZATION

We now present ParetoPrompt (Fig. 1), for prompt optimization that aims to cover the corresponding Pareto front of multi-objective NLP tasks with pre-trained LLMs.

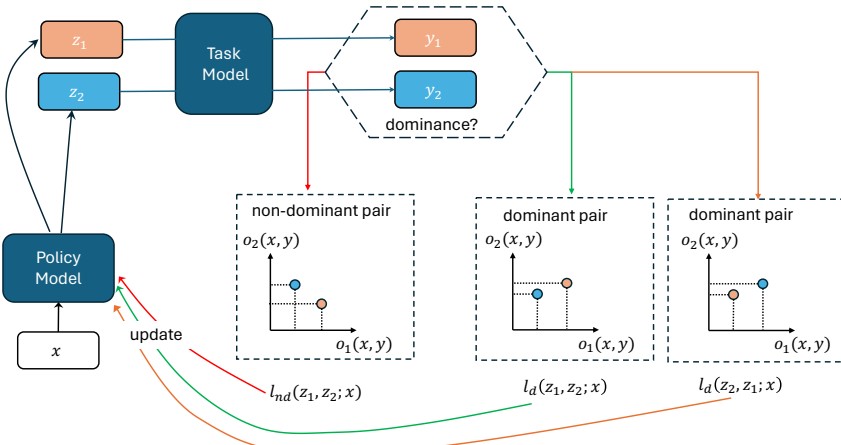

Figure 1: ParetoPrompt Iteration: ParetoPrompt trains a policy model to generate diverse and "Pareto optimal" prompts for an instance $x$. In each iteration, the policy model samples a pair of prompts. Based on the dominance relationships of their outputs, it calculates either the dominant or non-dominant loss, which is used to update policy model.

## 3.1 MULTI-OBJECTIVE PROMPT OPTIMIZATION

We consider the problem of generating Pareto optimal prompts to improve desired responses from a task-specific LLM, denoted as $T$. Given an input instance $x$, which can be reviews, queries or code, our objective is to design the prompts to guide the task-specific LLM towards generating high-quality responses $y$. Achieving high quality response $y$ can involve multiple, sometimes conflicting, objectives. Denote these objectives as $O(x, y) = [o_1(x, y), ..., o_n(x, y)]$, where each element, $o_i(x, y)$

represents a specific quality metric, such as relevance, fluency and/or creativity. These metrics can be automatically estimated using various techniques, for example based on the BLEU Score, Grammatical Error Rate, and Semantic Distance, or evaluated by another LLM (Zhang et al., 2023).

We aim to train a policy model $\pi_\theta$, using RL to generate prompts that approach the Pareto front. The policy model takes input $x$ and generates corresponding prompts $z$ with the probability $\pi_\theta(z|x)$. Here the prompt $z$ refers to the text provided to the task-specific LLM to generate the response $y = T(z)$. This includes both the instruction or query and any context provided before the the model's generated output. The simplest form of a prompt involves adding prefix text tokens in front of the input $x$ (Deng et al., 2022). A prompt $z$ is considered "Pareto optimal" if there exists no other prompt that can improve one objective in $O(x, y)$ without degrading at least one other objective. Mathematically, for two prompts $z_1$ and $z_2$, if $O(x, T(z_1)) \preceq O(x, T(z_2))$, and $O(x, T(z_1)) \neq O(x, T(z_2))$, then $z_1$ is "Pareto dominated" by $z_2$. Therefore, a prompt is "Pareto optimal" if it is not dominated by any other prompt. Finding such prompts allows us to obtain the best trade-off prompts considering multiple conflicting objectives.

The Pareto front represents the set of all Pareto optimal prompts, i.e. the set of prompts that "Pareto dominate" all the other prompts but mutually incomparable. In guiding the search for the Pareto front by RL training of the policy model, a common approach is to use a scalarization function that transforms the multi-objective problem into a standard single-objective problem. While convenient for training, these functions can impose oversimplified assumptions about the objective structure and limit the exploration of the entire Pareto front. The most common function is the weighted sum function, which prioritizes the solutions that maximize a pre-defined linear combination of objectives, neglecting potentially valuable parts of the Pareto front. Similarly, the hypervolume indicator (Zitzler et al., 2001), while rewards improvement in all objectives simultaneously, it may prioritize points that contribute most to the overall hypervolume increase, potentially neglecting areas with smaller hypervolume contributions. Furthermore, the preference of hypervolume may vary based on the choice of the reference point, introducing bias in the search process (Ishibuchi et al., 2017). Consequently, although using a sclarization function provides guidance in approaching the Pareto front in RL, it can also introduce bias in the search, limiting the algorithm's capability to cover the whole Pareto front.

In the following subsections, we introduce an innovative reward function for multi-objective prompt optimization. This approach leverages dominance relationships between prompts to guide the policy model towards the Pareto front. By doing so, we eliminate the need for predefined preferences imposed by traditional scalarization functions, allowing for a more flexible and comprehensive exploration of the solution space.

### 3.2 Dominance Preference-based Loss Function

Given an input instance $x$, we consider the dominance preference between two prompts $z_w$ and $z_l$ based on their corresponding outputs $y_w$ and $y_l$. Specifically, if $O(x, y_w) \succeq O(x, y_l)$, we define $z_w$ to be dominance preferable prompt over $z_l$, we denote as $z_w \succeq z_l$ for brevity. We define a data pair $(x, z_w, z_l)$ as dominance preference data, where $z_w$ dominates $z_l$ in terms of the resulting outputs. This data serves as the foundation for directly learning the policy model that generates Pareto optimal prompts.

Rafailov et al. (2024) introduced Direct Preference Optimization (DPO), a method that updates the policy model based on preference data without training a separate reward model. Since policy model $\pi_\theta$ is guided by the reward function, they showed that the reward function is implicitly connected to the policy model. This connection can be expressed as $r_\theta(x, z) \propto \log \frac{\pi_\theta(z|x)}{\pi_{\text{ref}}(z|x)}$, where $\pi_{\text{ref}}$ denotes a reference model. This reference model serves as a baseline or starting point for learning preferences, and it is typically chosen as an initialized model or a pre-trained model.

DPO utilizes a loss function derived from the Bradley-Terry model for preference modeling:

$$l_{\text{DPO}}(z_w, z_l; x) = -\log \sigma\left[\beta h(z_w, z_l; x)\right], \tag{1}$$

where $\beta$ is a scaling hyperparameter, $\sigma$ is the logistic sigmoid function and $h$ is the reward difference between $z_w$ and $z_l$ as defined:

$$h(z_w, z_l; x) = \log \frac{\pi_\theta(z_w|x)}{\pi_{\text{ref}}(z_w|x)} - \log \frac{\pi_\theta(z_l|x)}{\pi_{\text{ref}}(z_l|x)}. \tag{2}$$

This loss function $l_{\text{DPO}}$ increases the reward for the dominating prompt $z_w$ and decreases the reward for the dominated prompt $z_l$, thereby promoting the generation of $z_w$.

Extended from DPO, Identity Preference Optimization (IPO) (Azar et al., 2024) replaces the non-decreasing function related to the Bradley-Terry model by the identity function, resulting in a simpler loss function:

$$l_{\text{IPO}}(z_w, z_l; x) = \left[ h(z_w, z_l; x) - \frac{\tau^{-1}}{2} \right]^2, \tag{3}$$

where $\tau$ is a regularization hyperparameter, controlling $h(z_w, z_l; x)$ the reward difference between $z_w$ and $z_l$. DPO aims to maximize the difference to infinity, leading to overfitting to the preference dataset. In contrast, IPO aims to make the difference close to $\frac{\tau^{-1}}{2}$, therefore mitigating the risk of overfitting. The comparison of $l_{\text{DPO}}$ and $l_{\text{IPO}}$ as functions of $h(z_w, z_l; x)$ is shown in Fig. 2.

Both DPO and IPO losses utilize dominance preference data to update the policy model, aligning prompt generation with the dominance relationship between outputs. Note that dominance preference data is independent of specific objective values, so it remains robust against scaling or monotonic transformation of objectives.

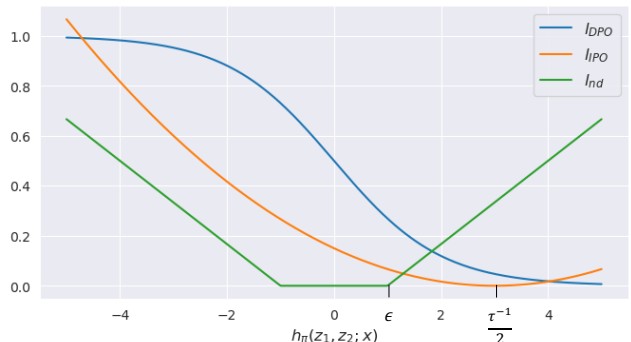

Figure 2: Comparison of loss functions, where $l_{\text{DPO}}$ and $l_{\text{IPO}}$ increase $h(z_w, z_l; x)$, the reward gap between dominant and dominated prompt pairs; while $l_{\text{nd}}$ reduces the reward gap between non-dominant pairs to near zero.

### 3.3 NON-DOMINATED LOSS FUNCTION

Based on preference update with dominance preference data, we are able to generate Pareto optimal prompts. However, there is no mechanism guaranteeing the coverage of the entire Pareto front. We therefore propose a non-dominated loss function based on the non-dominated data to encourage the policy model to generate diverse prompts to explore different trade-offs on the entire Pareto front.

Given a data pair $(x, z_1, z_2)$, it is possible that the corresponding outputs $y_1$ and $y_2$ are mutually non-dominated (incomparable), which we denote $z_1 \sim z_2$ and call $(x, z_1, z_2)$ non-dominated data. For these cases we do not expect a strong preference for either prompt. We want the policy model to assign similar likelihood in generating them. However, there should still be some tolerance for small likelihood differences between non-dominated prompts; otherwise, the loss function would force the model to assign identical likelihoods to all prompts. Therefore, we define a loss function that penalizes large differences in the rewards of $z_1$ and $z_2$, but tolerates small difference for non-dominated prompt pairs:

$$l_{\text{nd}}(z_1, z_2; x) = \lambda \max(|h(z_1, z_2; x)| - \epsilon, 0), \tag{4}$$

where $\lambda$ is a scaling hyperparameter, and $\epsilon$ is the tolerance of the difference between the reward function values. Note that as $|h(z_1, z_2; x)| = \left| \log \frac{\pi_\theta(z_1|x)}{\pi_\theta(z_2|x)} - \log \frac{\pi_{\text{ref}}(z_1|x)}{\pi_{\text{ref}}(z_2|x)} \right|$, the reward function difference actually reflects the deviation between the policy model $\pi_\theta$ and the reference model $\pi_{\text{ref}}$ in generating non-dominated prompts. Therefore, this loss function only becomes effective after the policy updates lead to the difference exceeding the threshold $\epsilon$. This ensures that the policy prioritizes learning dominant prompts first before focusing on diversifying non-dominated prompts.

The curve of $l_{\text{nd}}$ is shown in Fig. 2. From the comparison with $l_{\text{IPO}}$, we can conclude that when using $l_{\text{IPO}}$ and $l_{\text{nd}}$ as loss function terms, it's important to set $\epsilon < \frac{\tau^{-1}}{2}$ for hyperparameter selection to let $l_{\text{nd}}$ take effect.

### 3.4 PARETOPROMPT ALGORITHM

We now describe ParetoPrompt, our proposed training algorithm, for policy model generating Pareto-optimal prompts in detail.

The policy model leverages a pre-trained generative LLM with its latent embedding layers kept frozen. To fine-tune the model for prompt generation, a residual adapter module, implemented as a Multi-Layer Perceptron (MLP), is inserted between the latent layers and the model head. The reference model is set to be identical to the initial state of the policy model.

Briefly, we denote either the DPO or IPO loss function as $l_d$, then the overall loss function for ParetoPrompt is:

$$\mathcal{L}(\pi_\theta; \pi_{\text{ref}}) = \mathbb{E}_{(z_1, z_2; x) \sim \pi_{\text{ref}}} \big[ l_d(z_1, z_2; x) 1(z1 \succeq z2) + l_d(z_2, z_1; x) 1(z1 \preceq z2)$$
$$+ l_{nd}(z_1, z_2; x) 1(z_1 \sim z_2) \big]. \tag{5}$$

The training process is as follows (Fig. 1):

1. Randomly sample a training instance $x$.
2. Use the reference model to generate a pair of prompts, $z_1$ and $z_2$ for $x$.
3. Estimate the objectives of the corresponding outputs $y_1$ and $y_2$ and determine their dominance relationship, then select the either $l_d$ or $l_{nd}$ as loss function based on the dominance.
4. Use a gradient-based optimization algorithm (e.g., Adam) to update the policy model parameters $\theta$ based on the calculated loss.
5. Periodically update the reference model to match the current state of the policy model. This ensures that the reference model can leverage improved prompts for training as the policy model evolves.
6. Repeat above steps for a specified number of training iterations.

Advantage of our algorithm is that by combining the dominance preference loss function with non-dominated loss function, we encourage the policy model to generate Pareto optimal prompts while diversify the generation to explore the entire Pareto front. While ParetoPrompt avoids making assumptions about the multi-objective structure by not using a scalarization function, the way that ParetoPrompt currently treats non-dominant pairs cannot provide guide for generating better prompts as no preferences needs to be learned from them. However, in multi-objective problems, dominant pairs become rarer and non-dominant pairs become more frequent, especially with the increasing number of objectives, as in "many-objective' problems. As a result, ParetoPrompt can be inefficient in these scenarios.

## 4 EXPERIMENTS

To validate performance of our proposed PraretoPrompt, we apply it on both classification and text generation tasks, with multiple objectives. We also provide analyses of the ParetoPrompt algorithms.

**Baselines.** We compare our proposed ParetoPrompt with the following baselines:

1. **Summation** (Deng et al., 2022): This RL-based algorithm uses the reward function defined by scalarization, simply as the summation of different objectives: $r(x, y) = \sum o_i(x, y)$.
2. **Product**: This RL-based algorithm defines the reward as the product of different objectives:: $r(x, y) = \Pi o_i(x, y)$.
3. **HVI**: The algorithm uses the HyperVolume Increment (HVI) as the reward function within a RL framework. It tracks the Pareto front during training and use the hypervolume increment bring by each prompt as the reward.
4. **Reward-Guided IPO (R-IPO)**: This preference-based RL algorithm calculates the sum of objectives, and determines the preference of prompt pairs based on the summation, then updates the policy model using IPO loss in equation 3.
5. **InstOptima** (Yang & Li, 2023): This evolutionary algorithm utilizes the NSGA-II framework for multi-objective optimization. The mutation and crossover operators for prompts are executed using an LLM with corresponding operation prompts. In our experiments, we employ LLaMa 2 (7B) for prompt operations.
6. **ParetoPrompt DPO/IPO (PP-DPO/IPO)**: Our proposed ParetoPrompt algorithms.

## 4.1 FEW-SHOT TEXT CLASSIFICATION (TWO-OBJECTIVE TASK)

We conduct experiments on single-sentence classification across various datasets using token infilling with a BERT model (Brown, 2020). Classification is based on the probability of tokens corresponding to a set of verbalizers as class labels. We follow the prompt template `[Input][Prompt][Class]` as in Deng et al. (2022) and select the verbalizer token with the highest predictive probability at the `[Class]` position. For few-shot classification, We only take a small number of training samples and search for better prompts. We conduct experiments on a diverse set of popular few-shot classification tasks, including MR (Pang & Lee, 2005), SST-5 (Socher et al., 2013), Yelp-5 and Yahoo (Zhang et al., 2015).

**Objectives and Misaligned Metrics** We define a two-objective prompt optimization problem: besides optimizing the accuracy of the classification task, we also aim to optimize the fluency of the prompts. Various metrics can be used to evaluate the fluency. In this set of experiments, we use the grammatical acceptability score of RoBERTa-based-CoLA, a RoBERTa model fine-tuned for the Corpus of Linguistic Acceptability (CoLA) task (Morris et al., 2020). We denote the score as the *CoLA* score. This model is trained to classify whether a sentence is grammatically correct or not, and we aim to maximize the *CoLA* score to generate fluent prompts.

Since fluency can also be quantified by the perplexity of text as calculated by a language model, to demonstrate the robustness of ParetoPrompt against potentiallly misaligned metrics, we employ two distinct training signals: the CoLA score and, in a separate series of experiments, the *perplexity* calculated using GPT-2, while consistently take CoLA scores as the true objective in the testing stage. We have analyzed the CoLA and Perplexity scores of the prompts used in our experiments, and the results show a non-linear relationship between the CoLA scores and the perplexity scores. This non-linear relationship leads to poor performance when RL algorithms, trained using specific perplexity scores as a reward signal, while evaluated using CoLA scores. Details of the correlation analysis can be found in Appendix A.2.

**Experimental details** We use RoBERTa-large (Liu et al., 2021) as the LM for classification, while our policy model is based on DistilGPT2 (Sanh et al., 2019) with a two-layer MLP adaptor added before the head layer. The prompt search space consists of 5 discrete tokens. For all datasets, we randomly sample 16 samples per class for both the training and validation sets. The final performance is evaluated using a sufficiently large test set. For all RL-based algorithms (excluding InstOptima), 16 prompts are sampled for each iteration to calculate reward functions. Algorithms using dominance relationships (R-IPO and PP-DPO/IPO) employ 8 prompt comparison pairs for reward function calculation. Therefore, during the training, the total number of task language model queries is: $16 \times \text{class\_num} \times 16 \times 6{,}000$. Hyperparameters for the loss functions in equations (1), (3) and (4) are set as $\beta = 0.5$, $\tau = 0.5$, $\lambda = 1$ and $\epsilon = 0.1$. Each RL algorithm runs for 6K iterations for training. For InstOptima, we intialized with a population of 16 manually designed prompts and execute 60 generations of NSGA-II, matching the computational time of the RL algorithms. During the test stage, we generate 64 prompts for each dataset to perform a comprehensive comparison in terms of multi-objective performances. We conduct five independent runs to obtain average performance measures.

**Experimental Results** We use Hypervolume (HV) to evaluate the multi-objective performance of classification Accuracy and the prompt CoLA score, with the reference point set at (0, 0). HV for experiments using the CoLA score as the training signal is denoted as C-HV, while HV for experiments using Perplexity as training signal is denoted as P-HV. The difference between the two metrics, Diff-HV, measures the robustness of algorithm performance across these two settings. Based on the experimental results shown in Table 1, ParetoPrompt algorithms (IPO and DPO) consistently demonstrate superior performance across both C-HV and P-HV and have a small Diff-HV. In contrast, the Diff-HV for algorithms using scalarization rewards (Summation, Product and HVI) is significantly larger than other algorithms. That is due to the non-linear relationship between CoLA score and perplexity not only changes the absolute reward value, but also the relative rankings among prompts, leading to degraded performance. While the dominance relationships utilized by the ParetoPrompt algorithms remain unaffected by such transformations. Notably, InstOptima shows a small Diff-HV, partially due to its usage of NSGA-II, which relies on ranking based on dominance relationship.

Table 1: Comparison of dominated hypervolume (HV) by different methods in the bi-objective space defined by classification accuracy and CoLA score. "C-HV" refers to using the CoLA score as the training signal, and "P-HV" indicates using perplexity scores as training signal. The term "Diff-HV" represents the hypervolume difference between these two metrics. Higher values of "C-HV" and "P-HV" are preferable, while "Diff-HV" close to 0 indicates robustness against metric change.

| Dataset | Metric | Summation | Product | HVI | InstOpt | R-IPO | PP-DPO | PP-IPO |
|---------|--------|-----------|---------|-----|---------|-------|--------|--------|
| MR | C-HV | 76.5(8.4) | 63.3(5.2) | 77.2(6.9) | 70.6(9.1) | 75.5(8.5) | 80.9(4.3) | **83.0(4.6)** |
| | P-HV | 54.4(11.1) | 46.5(11.2) | 55.3(12.0) | 67.2(0.1) | 61.0(5.1) | 66.8(11.5) | **68.3(7.1)** |
| | Diff-HV | 22.1 | 16.8 | 21.9 | **3.4** | 14.5 | 14.1 | 14.7 |
| SST-5 | C-HV | 32.0(2.1) | 27.4(1.3) | **36.2(0.8)** | 31(3.7) | 29.5(7.0) | 34.1(2.5) | 35.3(0.5) |
| | P-HV | 23.6(3.1) | 20.4(1.2) | 24.4(2.5) | 32.6(1.9) | 29.9(1.3) | **34.3(0.4)** | 34.0(0.5) |
| | Diff-HV | 8.4 | 7.0 | 11.8 | -1.6 | -0.4 | -0.2 | 1.3 |
| Yahoo | C-HV | 39.8(1.9) | 27.5(1.2) | 42.2(2.6) | 40.7(3.9) | 24.9(1.3) | 47.1(1.7) | **48.2(0.5)** |
| | P-HV | 21.2(4.8) | 17.0(7.4) | 28.3 5.4 | 31.5(4.2) | 20.6(4.3) | 31.3(5.2) | **35.5(2.7)** |
| | Diff-HV | 18.6 | 10.5 | 13.9 | 9.2 | **4.3** | 15.8 | 12.7 |
| Yelp-5 | C-HV | 35.5(2.8) | 27.0(6.7) | 40.4(2.6) | 24.1(1.4) | 40.0(3.7) | **41.3(3.1)** | 40.6(2.1) |
| | P-HV | 21.9(2.1) | 19.7(4.8) | 20.5(3.1) | 35.1(2.5) | 37.1(4.4) | 39.7(2.5) | **39.8(2.3)** |
| | Diff-HV | 13.6 | 7.3 | 19.9 | -11.0 | 2.9 | 1.6 | **0.8** |

## 4.2 TEXT STYLE TRANSFER (THREE-OBJECTIVE TASK)

Following RLPrompt (Deng et al., 2022), we evaluate ParetoPrompt on a unsupervised text style transform task. The goal is to rewrite an input sentence into a desired style while still keep the content similar. Two conflicting objectives are considered: *style score* and *content similarity*. We also include include fluency of the prompts as a third objective. We conduct the task using the Yelp sentiment dataset (Shen et al., 2017) to convert Yelp negative reviews into positive ones while maintaining the content similarity. For example, the sentence"i will never be back" might be transformed into "i will be back again". The dataset consists of Yelp restaurant reviews, labelled by star ratings, with three or above as positive and those below three as negative. We randomly select 50 negative reviews for training, 50 for evaluation, and a separate set of 100 for testing.

**Objective Settings.** We adopt two model-based metrics for content similarity and sentiment positiveness evaluation. We set the content similarity objective as the content preservation reward function with the Compression, Transduction, and Creation (CTC) metric introduced in Deng et al. (2022), which measures the embedding alignment between the input and output. The sentiment objective is defined as the sentiment probability calculated using a BERT-based classifier fine-tuned on the Yelp dataset. For fluency objective, we kept using the CoLA scores calculated by the RoBERTa-based-CoLA model. Compared with few shot classification, this objective setting is more challenging for prompt optimization. In few shot classification, outputs are chosen among verbalizer, while text generation introduces greater output uncertainty. As a result, the quality of the objective signal is noisier, making it more difficult to identify prompts that achieve good average performance. To account for generative model randomness, we generate 128 outputs per prompt, averaging objective values for robust prompt evaluation.

**Experimental Details.** We use GPT-2 XL as the LM for the style transform task, while the prompt generation setting is the same as previous experiments. For all algorithms except InstOptima, during the training stage, each iteration processes a minibatch size of two input negative reviews, and four prompts are sampled for each negative review, which are then used to transform the inputs into positive reviews. Each algorithm runs for 10K iterations for training, resulting in a total number of language model queries equal to $128 \times 8 \times 10,000$. For InstOptima, we intialized with a population of 16 manually designed prompts and run 130 generations of NSGA-II, ensuring a comparable running time with the RL algorithms. During the testing stage, we again generate 64 prompts for each instance to evaluate the multi-objective performance. To ensure robustness, we conduct three independent runs to obtain an average performance measure.

**Experimental Results** The performance of generated prompts in the multi-objective space of a single run is shown in Fig. 3, providing an intuitive illustration. Additionally, Table 2 provides the average performance across three runs with mean and standard deviation values, with all objectives

set to maximize. Each algorithm generate 64 prompts from the policy model, and the Pareto Set Size represents the number of non-dominated prompts generated by each algorithm reflecting its ability to generate effective prompts. Two other metrics are evaluated for the overall performance in the multi-objective space: the Dominated HyperVolume (HV) and the Inverse Generation Distance (IGD). HV is the volume of the dominated region in the objective space with respect to the reference point $(0, 0, 0)$. IGD originally measures the average distance between the true Pareto front to the closest points in the objective space corresponding to generated prompts by different algorithms. A lower IGD value implies that the generated prompts achieve the performances closer to the Pareto front. Here since the ground-truth Pareto front is unknown, the Pareto optimal prompts of the combination of all the prompts act as the reference Pareto front.

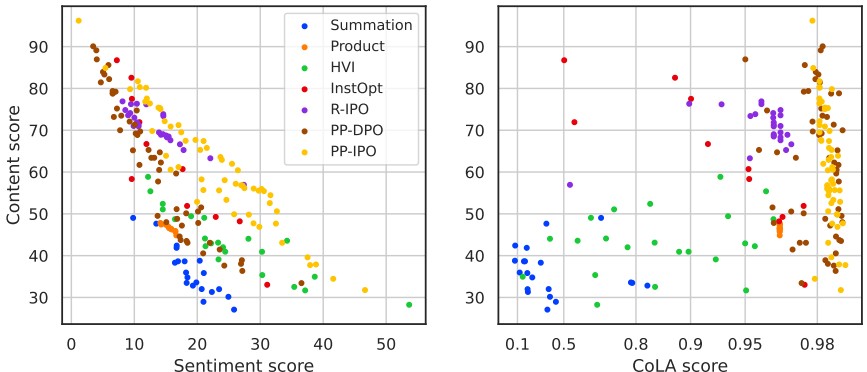

Figure 3: Visualization of three-objective prompt optimization results: Comparison of generated prompts from different algorithms in the objective space, with sentiment, content similarity, and CoLA score as objectives to be maximized. Results show that our ParetoPrompt algorithms effectively explore the entire Pareto front.

Table 2: Performance Comparison for Unsupervised Sentiment Transform. Pareto Set Size represents the number of non-dominated prompts, HV measures Dominated HyperVolume, IGD indicates distance to the Pareto front.

| Metric | Summation | Product | HVI | InstOpt | R-IPO | PP-DPO | PP-IPO |
|---|---|---|---|---|---|---|---|
| Max Style ↑ | 29.8(2.6) | 17.2(2.0) | 33.4(6.2) | 30.5(2.4) | 19.6(2.8) | 30.7(2.9) | **36.5(5.7)** |
| Max Content ↑ | 46.7(3.0) | 44.7(3.4) | 52.6(4.7) | 77.5(11.4) | 68.0(16.6) | 69.9(13.2) | **82.9(17.4)** |
| Max CoLA ↑ | 0.64(0.17) | 0.95(0.01) | 0.95(0.01) | 0.96(0.01) | 0.96 (0.01) | **0.98(0.0)** | **0.98(0.0)** |
| Pareto Set Size ↑ | 14.7(5.0) | 6.6(3.1) | 13.2(4.3) | 11.6 (3.2) | 26.8(5.4) | 34.0(3.6) | **34.8(10.9)** |
| HV ↑ | 572(135) | 725(45) | 1429(153) | 1838 (228) | 1166(241) | 2007(242) | **2406(361)** |
| IGD ↓ | 31.6(2.5) | 33.7(2.8) | 17.9(9.8) | 17.1(3.7) | 22.6(3.9) | 15.7(7.6) | **14.4(4.8)** |

From Fig. 3, we observe that Summation, Product and HVI do not perform well due to the fact that the inaccurate reward cannot guide the search effectively. R-IPO tends to form clusters because it lacks a mechanism to generate diverse prompts but simply exploits the preference information provided by preference data. In contrast, our proposed ParetoPrompt algorithms (especially PP-IPO) can cover the entire Pareto front due to the introduction of non-dominance loss. Table 2 shows that our proposed ParetoPrompt algorithms achieve higher HV and smaller IGD, reflecting that the prompts generated by Pareto-Prompt effectively cover the whole Pareto front. Additionally, we observe that PP-IPO performs better than PP-DPO, probably because the

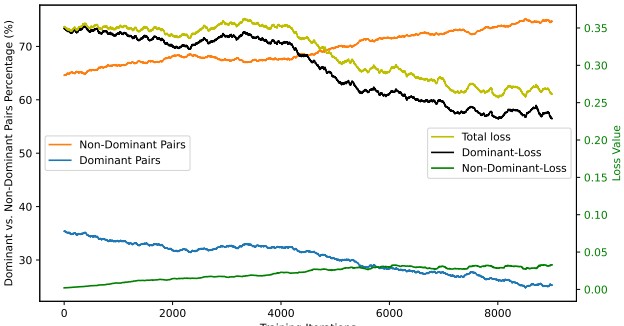

Figure 4: Training Analysis of ParetoPrompt-IPO: Trends in dominant vs. non-dominant prompt pair percentages, and total, dominance, and non-dominance loss changes.

IPO loss does not suffer from the potential overfitting issues associated with DPO, which allows PP-IPO to explore the search space more efficiently. Overall, our ParetoPrompt algorithms demonstrate superior performances by effectively covering the entire Pareto front with diverse prompts.

### 4.3 PARETOPROMPT TRAINING ANALYSIS

We analyze the ParetoPrompt algorithms during the training process, specifically focusing on the text style transfer task. In the ParetoPrompt-IPO algorithm, we present the percentage change of dominant and non-dominant prompt pairs in Fig. 4. Additionally, we include the non-dominated loss defined in Eq. (5). The figure reveals that as training progresses, the algorithm samples more non-dominant prompt pairs, indicating it learns to generate trade-off prompts. The non-dominated loss starts at a value of 0 and increases as training continues. This increasing non-dominated loss suggests that as the algorithm approaches the Pareto front, it focuses more on exploring non-dominated prompts. Our experimental results shown in Appendix A.3 also indicate that without the non-dominated loss, the dominate-only algorithm tends to generate prompts in clusters.

## 5 CONCLUSION

We have developed ParetoPrompt, a RL-based prompt optimization algorithm for multi-objective text generation. The algorithm's training relies solely on the multi-objective dominance relationships between pairs of prompts, and requires no predefined scalarization function, thus allows us bypass assumptions about human preferences in text evaluation. ParetoPrompt defines separate loss functions for dominant and non-dominant prompt pairs. The combination effect of these loss functions encourages the generation of Pareto-optimal prompts but diversifies the prompts to cover the entire Pareto front. Moreover, by using only dominant relationships, the algorithm performs robustly even when there is a mismatch between training and testing metrics. Additionally, it can incorporate preference data for training. Overall, ParetoPrompt presents a preference learning approach for generating Pareto-optimal prompts, providing a promising direction for multi-objective prompt optimization.

## 6 LIMITATION AND FUTURE WORK

As highlighted in Section 4.3, ParetoPrompt may become inefficient when addressing problems with a large number of objectives. To address this limitation, future work can explore incorporating relaxed Pareto dominance relations (López Jaimes & Coello Coello, 2009), which extends the definition of dominance to capture subtle preference information between non-dominant prompts. Examples include the $(1-k)$-dominance relation (Farina & Amato, 2002) and the expansion relation that controls the dominance area of solutions (Sato et al., 2007). By incorporating these relaxed relations, we can potentially use the information from non-dominant pairs to guide the policy model's updates in many-objective cases.

### ACKNOWLEDGEMENT

This work was conducted as part of the LUCID (Low-dose Understanding, Cellular Insights, and Molecular Discoveries) program, supported by the U.S. Department of Energy, Office of Science, Office of Biological and Environmental Research, under Contract DE-AC02-06CH11357. X.Q. acknowledges partial support from National Science Foundation (NSF) through grants SHF-2215573 and IIS-2212419.

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

# A   APPENDIX

This appendix presents supplementary materials, including the pseudo-code for the ParetoPrompt algorithm, an analysis of CoLA and perplexity scores, and results from two-objective prompt optimization experiments in text style transfer.

## A.1   PSEUDO-CODE FOR PARETOPROMPT

Pseudo-code for ParetoPrompt is summarized in **Algorithm 1**.

---
**Algorithm 1** ParetoPrompt

---
**Require:** Training dataset $\mathcal{X}$, Reference model $\pi_{\text{ref}}$, Loss Hyperparameters
 1: Initialize policy model $\pi_\theta \leftarrow \pi_{\text{ref}}$
 2: **for** epoch in range(num_epochs) **do**
 3:     **for** $x$ in $\mathcal{X}$ **do**
 4:         **if** $z1 \succeq z2$ **then** loss $= l_d(z_1, z_2; x)$
 5:         **else if** $z1 \preceq z2$ **then** loss $= l_d(z_2, z_1; x)$
 6:         **else** loss $= l_{nd}(z_1, z_2; x)$
 7:         **end if**
 8:         Update $\pi_\theta$ with gradient descent on loss
 9:     **end for**
10:     **if** (epoch% update_period) $== 0$ **then** $\pi_{\text{ref}} \leftarrow \pi_\theta$
11:     **end if**
12: **end for**

---

## A.2   ANALYSIS OF COLA AND PERPLEXITY SCORES

We analyzed the CoLA and Perplexity scores of the prompts used in our experiments. We generated 1000 prompts of 5 tokens randomly and calculate their CoLA score and perplexity. And their relationship is shown in Fig. S1. Notably, the CoLA score exhibits an approximately linear relationship with log perplexity.

The Spearman's Rank Correlation Coefficient between them is -0.473, indicating a moderate negative correlation. In contrast, the linear correlation coefficient is -0.253, reflecting a weak negative linear relationship. However, when we apply a logarithmic transformation to Perplexity, the linear correlation coefficient increases to -0.471. This suggests a non-linear relationship between CoLA and Perplexity scores, which can be approximately described by a logarithmic transformation.

## A.3   TEXT STYLE TRANSFER WITH TWO OBJECTIVES

We also conduct two-objective prompt optimization experiments on the text style transfer task. The settings are the same as Sec. 4.2, except that we optimize prompts for two objectives: content similarity and sentiment positiveness. Fig. S2 provides an intuitive illustration of the generated prompts in the objective space, Fig. S3 also provides the average performance across five runs.

Notably, the results shows that without the non-dominance loss, the Dominance-Only algorithms tend to form clusters because of the lack of mechanism to diversify prompts, as shown in Fig. S2. In contrast, ParetoPrompt can cover the entire Pareto front due to the introduction of non-dominance loss. Fig. S3 shows that our proposed ParetoPrompt algorithms achieve higher D-HV with smaller variance because diverse prompts lead to higher and more robust D-HV. In contrast, the competing algorithms generate prompts that cluster together, causing the D-HV varies with the cluster's position and resulting in larger variance. The smaller IGD of ParetoPrompt reflects it has smaller distance from the reference Pareto front, reflecting that the prompts generated by ParetoPrompt effectively cover the whole Pareto front. Overall, our ParetoPrompt algorithm demonstrates superior performance by effectively covering the entire Pareto front with diverse prompts.

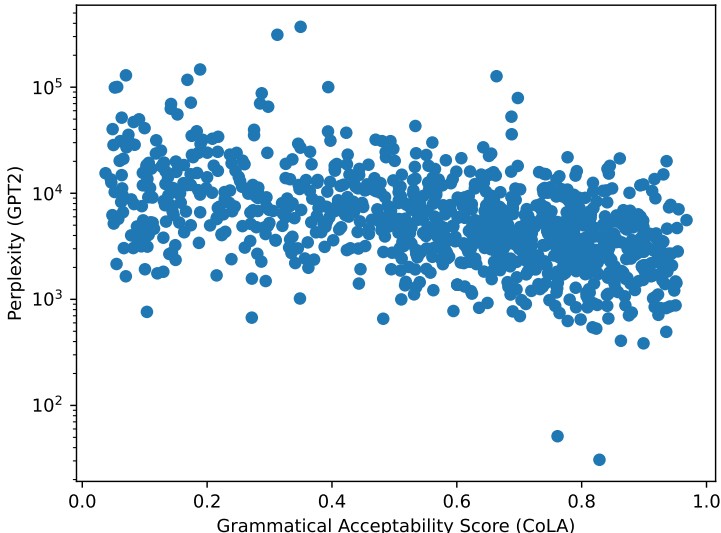

Figure S1: Relationship between CoLA and Perplexity scores.

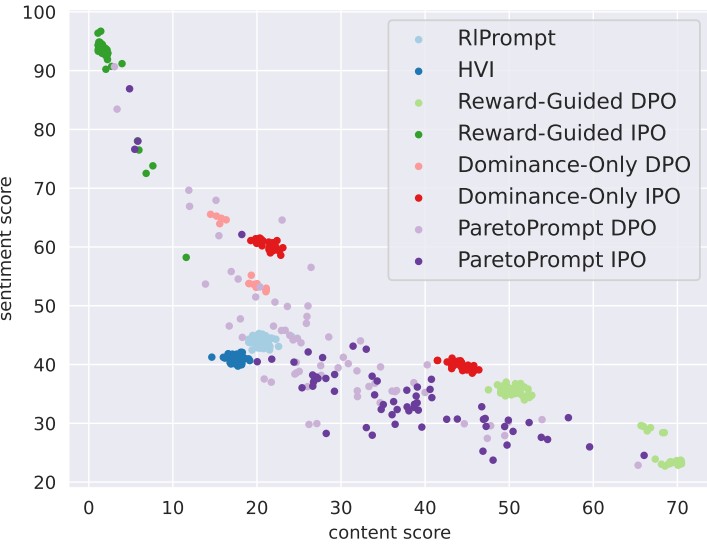

Figure S2: Bi-objective Prompt Optimization Illustration. Comparison of generated prompts from different algorithms for a single instance in bi-objective space, with sentiment and content similarity as the two objectives to be maximized. Results show that our proposed ParetoPrompt algorithms effectively explore the entire Pareto front.

## A.4 TEXT STYLE TRANSFER WITH FOUR OBJECTIVES

In this section, we present preliminary results from an extended experiment evaluating our method on a four-objective optimization task in text style transfer. While prior studies on multi-objective prompt optimization typically consider at most three objectives, our method is fully capable of handling more complex scenarios. To demonstrate this, we evaluate performance under the following four objectives: Style score, Content similarity, Output fluency and Conciseness. The first three objectives are the same as Sec. 4.2, while the fourth objective, Conciseness is measured by the length of the output.

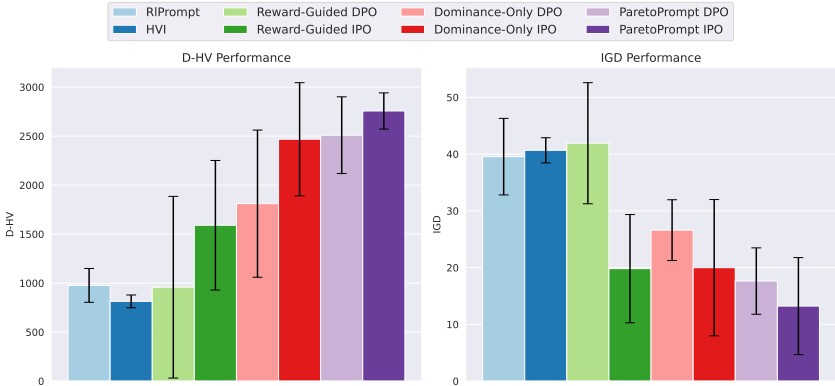

Figure S3: Performance Comparison for Unsupervised Sentiment Transform. We compares the performance of various algorithms in an unsupervised sentiment transform task. D-HV (higher is better) measure Dominated HyperVolume, IGD (lower is better) indicate distance to the Pareto front. Our ParetoPrompt surpasses competing algorithms in both metrics.

We present the available results for methods that could be easily adapted to this four-objective setting. These results provide insights into the capability of our method in handling a larger number of objectives. The results, based on three independent runs per method, are summarized in Table S1.

Table S1: Performance Comparison on Four-Objective Text Style Transfer. Pareto Set Size represents the number of non-dominated prompts, HV measures Dominated HyperVolume, and IGD indicates distance to the Pareto front.

| Metric | Summation | Product | R-IPO | PP-DPO | PP-IPO |
|---|---|---|---|---|---|
| HV ↑ | 6279 (634) | 6793 (659) | 16895 (575) | 23199 (170) | **30861 (154)** |
| IGD ↓ | 34.57 (1.35) | 34.18 (2.07) | 22.55 (2.55) | 8.98 (0.55) | **3.05 (0.44)** |
| Pareto Set Size ↑ | 15.67 (1.70) | 15.3 (0.47) | 25.5 (6.5) | 33.5 (1.5) | **51.5 (1.5)** |

These preliminary results highlight the ability of our method to optimize across multiple objectives.

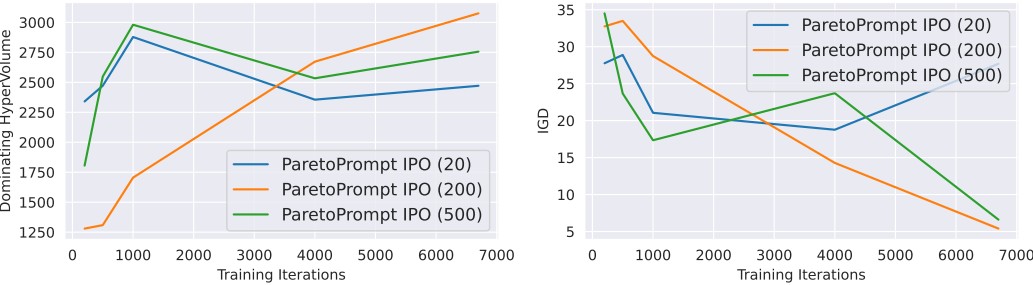

Figure S4: Ablation performance comparison with different update period choices.

## A.5 ABLATION: REFERENCE MODEL UPDATE PERIOD

The reference model update period (in **Algorithm 1** line 10) plays a crucial role in balancing the convergence speed and stability during training. With a short update period, the reference model closely tracks the current policy model, potentially leading to faster convergence; but such a choice can also lead to unstable training. In contrast, a long update period can lead to more stable training. However, the reference model may hinder the further improvement of the policy model. We conduct an ablation study with the different reference model update periods set to 20, 200 and 500. The results are shown in Fig. S4. From the figure, we can observe that a period of 200 achieves a good balance between convergence speed and performance in both D-HV and IGD. This setting allows the reference model to adapt sufficiently to guide the policy's learning while maintaining stability for exploration.

