# OpenReview forum: "Pareto Prompt Optimization"
_ICLR.cc/2025/Conference — ICLR 2025 Poster_

### Official Review · Reviewer_yPkv · 2024-11-01

**Soundness:** 2
**Presentation:** 3
**Contribution:** 3
**Rating:** 8
**Confidence:** 4

**Summary:**

The authors proposed a prompt optimization method relying on the pareto dominance relationships between prompts. This avoids a predefined aggregation among the multiple objectives. Instead, a loss containing separate components for dominance and non-dominance data is designed, optimizing for pareto optimal prompts on the pareto frontier. Experiments are conducted on real-world datasets and language models, showing the method’s capability of producing well-performing prompts across multiple objectives.

**Strengths:**

1. Studies an important practical problem of multi-objective metric design when the explicit structure is unknown.
2. The special treatment for non-dominated prompts contributes to the novel design of the overall loss function.
3. Clear visual illustration of the results lying on the pareto front.

**Weaknesses:**

1. The experiments are not extensive enough, e.g., no experiment beyond 3 objectives, and only models like BERT and GPT2 are used.
2. The experiment design on the choice of objectives can be improved. Having prompt fluency as the objective makes less sense as compared to output fluency, conciseness, informativeness, style alignment, etc.

**Questions:**

1. Since non-dominated prompts consist of the majority of the sampled prompts, will they also dominate the loss function? For example, if most of the prompts are non-dominated, optimizing the loss function basically gives the same reward for almost all prompts. Please clarify.
2. Clarify how having the loss in equation (4) encourages “diversifying non-dominated prompts” as stated in line 257?
3. In the experiments, the fluency of the prompt is used as an objective. This is counterintuitive: I would imagine the fluency of the output/generation to be much more important than the fluency of the prompt itself. This makes the experiment results less convincing.
4. From Figure 3, I can see that ParetoPrompt produces many prompts that lie on the “pareto front”. If I need to choose one of them eventually to use in the system, how should I choose the prompt?
5. It seems strange to me that the percentage of non-dominated samples increases with training, but the loss also keeps increasing.
6. When the number of objectives increases, it is expected to have much more conflicts (non-dominant pairs) as compared to dominant ones. Though this is discussed as a limitation in the last section, I always have this doubt in mind while reading the paper, do consider bringing this comment to an earlier section of the paper for clarity. On a side note, I would like to know the rough maximum number of objectives that ParetoPrompt can handle in practice?
7. A missing related work [1] on using preference optimization for prompt optimization.
8. [Typo] Line 404, “ouput” → “output”

References:

[1] Prompt Optimization with Human Feedback. In ICML 2024 Workshop on Models of Human Feedback for AI Alignment.

---

> ### Author Response · Authors · 2024-11-21
>
> ### 1. Objectives Selection and Additional Result
>
> In the current literature on multi-objective prompt optimization, reported benchmarks are typically limited to at most three objectives, which guides our choice to evaluate our method using two or three objectives for fair comparison with existing methods. We thank the reviewer for suggesting output fluency and conciseness as objectives and have extended our text style transfer experiment to include four objectives:
>
> 1. **Style score**
> 2. **Content similarity**
> 3. **Output fluency**
> 4. **Conciseness** (minimizing the length of the output)
>
> For each algorithm, we conducted three independent runs, and the average performance for the four-objective problem is provided below:
>
>
> | Method          | Summation      | Product       | R-IPO        | PP-DPO        | PP-IPO          |
> |------------------|----------------|---------------|--------------|---------------|-----------------|
> | **HV ↑**        | 6279 (634)     | 6793 (659)    | 16895 (575)  | 23199 (170)   | **30861 (154)** |
> | **IGD ↓**       | 34.57 (1.35)   | 34.18 (2.07)  | 22.55 (22.55)| 8.98 (0.55)   | **3.05 (0.44)** |
> | **Pareto Set Size ↑** | 15.67 (1.70) | 15.3 (0.47)   | 25.5 (6.5)   | 33.5 (1.5)    | **51.5 (1.5)**  |
>
> The full results will be included in the revised paper.
>
> ---
>
> ### 2. Model Selection
>
> Regarding our choice of BERT and GPT-2 as task models, we followed prior works such as [1] and [2], which also used these models in their evaluations. However, we recognize the importance of including more recent language models and will extend our experiments to incorporate them.
>
> ---
>
> ### Q1: Non-Dominance Loss
>
> The non-dominance loss does not **“govern”** the loss function, as shown in the updated Figure 5 in our revised manuscript. The reviewer is correct in observing that the majority of the sampled prompts are non-dominated. However, we designed the non-dominance loss to have little or even zero value when the reward difference (reflecting the likelihood differences) between the non-dominant pairs is small. When we update $π_θ$ to equal $π_{\text{ref}}$, the $h$ function in Eq. (4) becomes zero, and thus the non-dominance loss is zero. In this scenario, training is driven by the dominance loss, with the non-dominance loss acting as a form of regularization to encourage diversity in prompt generation.
>
> ---
>
> ### Q2: Loss Function in Eq. (4)
>
> The loss in Eq. (4) is based on the reward function $\log(\pi_\theta(z|x) / \pi_{\text{ref}}(z|x))$ from the DPO paper [2], which reflects the likelihood of generating a specific prompt $z$. In Eq. (4), we penalize large reward differences between non-dominant pairs. This encourages the model to distribute generation likelihoods more evenly across Pareto-optimal prompts. By preventing any single Pareto-optimal prompt from dominating the likelihood distribution, the model is able to better diversify and generate a broader set of non-dominated prompts.
>
> ---
>
> ### Q3: Additional Experiments
>
> We have conducted experiments taking output fluency as one of the four objectives, as described in the new experiment above.
>
> ---
>
> ### Q4: Prompt Selection
>
> To choose one prompt from the Pareto front to use in the system, the decision depends on the specific priorities or preferences for specific applications. For example, if output fluency is more critical than conciseness, we might prioritize prompts that perform better on fluency while slightly compromising on conciseness. The preferences can also be dynamic. Our algorithm provides a diverse set of Pareto-optimal prompts, with higher flexibility leaving the final choice to decision-makers based on their specific needs.
>
> ---
>
> ### Q5: Loss Dynamics
>
> We appreciate your detailed observation and good question. Although the non-dominance loss increases and plateaus during training, the total loss is decreasing overall. We have updated Figure 5 to include both the total loss and the dominated loss to provide a clearer picture. The non-dominated loss acts as a form of regularization, it encourages the model to maintain diversity and explore the Pareto front effectively, which can result in a slight increase in its value as the model generates more competitive prompts.

---

> > ### Author Response · Authors · 2024-11-21
> >
> > ### Q6: Limitations
> >
> > We thank the reviewer for the suggestion, we agree that highlighting this limitation earlier in the paper could improve clarity, we have mentioned this limitation in Section 3.4 of our updated manuscript.
> >
> > Regarding the maximum number of objectives that ParetoPrompt can handle, this largely depends on the specific problem and available computational budget. In a simplified setting where the objective space is a hypercube and the objectives are independent, the probability of generating dominant pairs is $\frac{1}{2^n}$​, where $n$ is the number of objectives. With each iteration generating at most 100 prompt pairs for comparison, the practical limit on the number of objectives is approximately 7, beyond which the dominance pairs become too rare, making the optimization less effective. For the cases with conflicting objectives, the maximum number of objectives should be even smaller.
> >
> >
> > ---
> >
> > ### Q7: Reference Paper
> >
> > We thank the reviewer for suggesting the reference paper. This paper relies on learning a reward model trained on human preference data and then optimizing the reward model to find the optimal prompt. We acknowledge the strengths of this reference paper in prompt optimization and have included it in the related works section of the updated manuscript.
> >
> > ---
> >
> > **Typo Corrections**
> >
> > We appreciate the reviewer pointing out the typo. We have tried our best to proofread the updated manuscript to avoid typos.
> >
> > ---
> >
> > **References**
> > [2] Rafailov, Rafael, et al. "Direct preference optimization: Your language model is secretly a reward model." *Advances in Neural Information Processing Systems 36* (2024).
> > [3] Deng, Mingkai, et al. "Rlprompt: Optimizing discrete text prompts with reinforcement learning." arXiv preprint arXiv:2205.12548 (2022).
> > [4] Jafari, Yasaman, et al. "MORL-Prompt: An Empirical Analysis of Multi-Objective Reinforcement Learning for Discrete Prompt Optimization." arXiv preprint arXiv:2402.11711 (2024).

---

> > > ### Comment · Reviewer_yPkv · 2024-11-22
> > >
> > > I thank the authors for the response. My concerns are addressed. I think this is a good paper overall, I will raise my score to reflect my positive opinions.

---

> > > > ### Author Response · Authors · 2024-11-24
> > > >
> > > > Dear Reviewer,
> > > >
> > > > Thank you for improving your rating and for acknowledging our rebuttal.

---

### Official Review · Reviewer_x6AH · 2024-11-04

**Soundness:** 3
**Presentation:** 4
**Contribution:** 2
**Rating:** 6
**Confidence:** 4

**Summary:**

This paper proposed a Pareto prompt optimization algorithm. Instead of considering a single objective, this paper considers multi-objective in a real prompt optimization scenario and proposes an RL-based algorithm to find the Pareto optimal prompts in the Pareto front and achieve better performance than other algorithms.

**Strengths:**

1. The Pareto prompt optimization problem proposed by the author is novel and practical for me
2. The presentation of the algorithm design is great
3. The experimental results shows that the Pareto prompt optimization algorithm outperforms other methods.

**Weaknesses:**

1. This paper did not provide enough discussion on the related works of prompt optimization, [1,2,3,4] are some of the works that I think should be included.

2. In the comparison, the author did not provide any comparison with existing prompt optimization works like [1,2,3,4]. More justification of comparison on this is needed to position this paper in the area of prompt optimization.

[1] Yang, C., Wang, X., Lu, Y., Liu, H., Le, Q. V., Zhou, D., & Chen, X. (2024). Large language models as optimizers. arXiv. https://arxiv.org/abs/2309.03409
[2] Fernando, C., Banarse, D., Michalewski, H., Osindero, S., & Rocktäschel, T. (2023). Promptbreeder: Self-referential self-improvement via prompt evolution. arXiv preprint arXiv:2309.16797.
[3] Lin, X., Wu, Z., Dai, Z., Hu, W., Shu, Y., Ng, S. K., ... & Low, B. K. H. (2024). Use your instinct: Instruction optimization using neural bandits coupled with transformers. ICML 2024.
[4] Guo, Q., Wang, R., Guo, J., Li, B., Song, K., Tan, X., ... & Yang, Y. (2023). Connecting large language models with evolutionary algorithms yields powerful prompt optimizers. arXiv preprint arXiv:2309.08532.

**Questions:**

1. How does the approach proposed in this work different from [5]? Since [5] also considers the human preference in prompt optimization (in this paper's case, dominance relationship). Could the author provide some explanation? If indeed, there are some similarity, is it possible to compare this work as one of the baseline in the paper?

[5] Lin, X., Dai, Z., Verma, A., Ng, S. K., Jaillet, P., & Low, B. K. H. (2024). Prompt Optimization with Human Feedback. arXiv preprint arXiv:2405.17346.

---

> ### Author Response · Authors · 2024-11-21
>
> Thank you for your thorough assessment of our submission and for your helpful recommendations.
>
> ---
>
> ### 1. Discussion and Comparison with Related Work
>
> We appreciate the reviewer’s suggestion to compare our method with the suggested reference papers. However, all the reference papers focus on **single-objective** prompt optimization, while our method is explicitly designed for **multi-objective** prompt optimization.  Specifically, references **[2]** and **[4]** employ evolutionary algorithms for single-objective prompt optimization and we have compared our method with their multi-objective counterpart, **InstOptima**, which uses **NSGA-II** for multi-objective prompt optimization.
>
> We acknowledge the strengths of the reference papers in single-objective prompt optimization and have included a discussion of their contributions in the **Related Works** section of our paper.
>
> ---
>
> ### 2. Differences and Comparison with Related Work
>
> We thank the reviewer for their question. While **[5]** also considers human preference in prompt optimization, it focuses on **single-objective** prompt optimization, whereas our method addresses **multi-objective** prompt optimization.
>
> Moreover, **[5]** relies on learning a **reward model** trained on human preference data and then optimizes the reward model to find the optimal prompt. In contrast, our approach directly optimizes the prompt generation model without requiring explicit reward modeling. This direct optimization makes our method both more efficient by eliminating the intermediate reward modeling step and more robust by reducing reliance on an optimization procedure.

---

> > ### Comment · Reviewer_x6AH · 2024-12-02
> >
> > Thank you for your response. I will keep my current score.

---

### Official Review · Reviewer_9fmx · 2024-11-09

**Soundness:** 3
**Presentation:** 3
**Contribution:** 2
**Rating:** 6
**Confidence:** 2

**Summary:**

This paper introduces ParetoPrompt, a reinforcement learning method designed for multi-objective prompt optimization.

**Strengths:**

The reported experiments show the proposed algorithm outperform the baseline methods under a variety of metrics

**Weaknesses:**

1. More recent baseline methods should be compared. For example: https://arxiv.org/abs/2406.12845
2. Any theoretical justifications that the proposed training process (in the end of section 3) is Pareto-optimal?
3. In the proposed training process, how to estimate the objectives of the corresponding outputs y1 and y2?

**Questions:**

see above

---

> ### Author Response · Authors · 2024-11-21
>
> Thank you for the thoughtful review and feedback on our paper.  Below are our responses to address the concerns and questions raised:
>
> ---
>
> ### 1. Suggested Baseline Paper
>
> We thank the reviewer for suggesting the baseline paper. The paper proposes a multi-objective reward modeling method trained on human preference data, where the multi-objective framework enhances both interpretability and steerability in RLHF fine-tuning.
>
> We agree that this method could be extended for prompt optimization by building a reward model for prompts and then optimizing the reward model to find the optimal prompt. However, this extension is not straightforward and would require significant adaptation, including:
> 1. **Training the reward model for prompts** in a domain- and application-specific manner.
> 2. **Devising an optimization procedure** compatible with the prompt space.
>
> While the approach is promising, incorporating it into our work within the given rebuttal time is not feasible and is beyond the current scope. We’ll consider including it in future studies to enrich the experimental comparisons.
>
> Additionally, we’d like to highlight that, compared to reward modeling approaches, our method directly optimizes the prompt generation model without the need for explicit reward modeling. This direct optimization makes our method both more efficient by eliminating the intermediate reward modeling step and more robust by reducing reliance on an optimization procedure.
>
>
> ---
>
> ### 2. Theoretical Proof of Pareto-Optimality
>
> We are currently not able to provide a theoretical proof that the proposed training process is Pareto-optimal, while we are actively working on it. Establishing such a proof is challenging due to the **inherent complexity** of multi-objective optimization in high-dimensional spaces and the **stochastic nature** of the training process.
>
> Specifically, the optimization landscape for multi-objective problems is typically non-convex, the trade-offs among objectives can be complex and task-dependent, making it difficult to derive guarantees for Pareto-optimality. Additionally, our method involves a stochastic prompt sampling process, which adds complexity to formalize a theoretical proof.
>
> Despite the lack of formal proof, our **empirical results** demonstrate that the method effectively approximates the Pareto front across various tasks, supporting its practical utility.
>
>
> ---
>
> ### 3. Estimate the Relative Preference
>
> Our training procedure requires only the **relative preference** between $y_1$ and $y_2$ for each objective, rather than their absolute values. This is simpler to obtain and more flexible in practical applications. The relative preference can be derived either through evaluations by a LLM or estimated using a surrogate measure function, For example, in our experiments, we used CoLA to estimate text fluency.

---

### Official Review · Reviewer_UXSm · 2024-11-11

**Soundness:** 3
**Presentation:** 2
**Contribution:** 3
**Rating:** 6
**Confidence:** 3

**Summary:**

This paper addresses the multi-objective prompt optimization challenge through a novel method called ParetoPrompt. In particular, ParetoPrompt introduces a reinforcement learning from human feedback (RLHF) approach that utilizes dominance preference data, enabling it to efficiently explore optimal trade-offs across multiple objectives. This approach is both innovative and promising, as it allows for nuanced optimization without relying on rigid scalarization functions.

**Strengths:**

- Introducing reinforcement learning from human feedback (RLHF) into Pareto optimization is a novel and inspiring approach, adding a valuable dimension to multi-objective optimization.
- The final results are promising, demonstrating the method's potential to achieve balanced and effective trade-offs across objectives.

**Weaknesses:**

- The motivation for using RLHF in Pareto optimization, as opposed to standard Pareto algorithms, could be further elaborated to strengthen the case for this approach.
- The paper lacks a detailed report on querying budgets (or optimization efficiency), which is critical for assessing practical performance in prompt optimization.
- The learning procedure for the reward model appears complex and may be challenging to implement or adapt across diverse applications.
- Including more results with a greater number of objective functions would enhance the evaluation and demonstrate broader applicability.

**Questions:**

See above.

---

> ### Author Response · Authors · 2024-11-21
>
> We sincerely thank the reviewer for their thoughtful feedback and valuable suggestions. We address the specific concerns and questions below:
>
> ---
>
> ### 1. Motivation for Using DPO (Direct Preference Optimization)
>
> The motivation for using DPO in Pareto optimization stems from two key considerations:
>
> - **Robustness to Evaluation Challenges:**
>   Standard Pareto algorithms often rely on scalarized or absolute objective values to guide optimization. However, in language generation tasks, accurately quantifying absolute objective values is often impractical due to vague evaluation criteria and subjective evaluations. DPO, by focusing on relative preferences derived from dominance relationships, provides a more robust approach that sidesteps the unreliability of absolute evaluations.
>
> - **Flexibility in Multi-Objective Assumption:**
>   Standard Pareto algorithms often require assumptions about the structure of the objective space, such as additive contributions, uniform preferences, or pre-defined reference points. These assumptions may not hold in complex, real-world problems like prompt optimization for language generation. By solely relying on dominance relationships, DPO avoids such biases, ensuring a more flexible and assumption-free optimization process.
>
> ---
>
> ### 2. Query Budget Details
>
> In our experiments, we use the same query budget across **Summation**, **Product**, **HVI**, **R-IPO**, **PP-DPO**, and **PP-IPO**. The query calculations for Tables 1 and 2 are as follows:
>
> - **Table 1:**
>   The policy model generates 16 prompts per iteration. For each prompt, the task model queries 16 samples per class. With a total of 6,000 iterations, the total number of language model queries is:
>   `16 × class_num × 16 × 6,000`
>
> - **Table 2:**
>   The policy model generates 8 prompts per iteration. Each prompt produces 128 style-transformed outputs for average performance evaluation. With 10,000 iterations, the total number of queries is:
>   `128 × 8 × 10,000`
>
> This consistent query budget ensures fairness when comparing methods in both tables.
>
> InstOptima, on the other hand, generates prompts through prompt operations such as mutation and crossover of parent prompts using **LLaMA2-7B**. To ensure a comparable total runtime with **PP-DPO/IPO**, we employed **NSGA-II** for prompt optimization, running:
> - 60 generations with 16 prompts per generation for Table 1
> - 130 generations with 16 prompts per generation for Table 2
>
> These details have been updated in our paper.
>
> ---
>
> ### 3. Generalization Across Diverse Tasks
>
> Our learning procedure is designed to be a general approach that can be applied across diverse tasks, provided that the dominance relationships data is available. As demonstrated in our experiments, the procedure successfully adapts to various tasks.
>
> ---
>
> ### 4. Additional Experimental Results with Four Objectives
>
> We thank the reviewer for the suggestion to include results with a greater number of objective functions. In the current literature on multi-objective prompt optimization, reported benchmarks are typically limited to at most three objectives, which guides our experiment setting with two or three objectives for fair comparison with existing methods.
>
> Our method is capable of handling problems with more objectives. To demonstrate this, we extended our text style transfer experiment to include four objectives:
>
> 1. **Style score**
> 2. **Content similarity**
> 3. **Output fluency** (as suggested by reviewer yPkv)
> 4. **Conciseness** (minimizing the length of the output)
>
> For each algorithm, we conducted three independent runs, and the average performance for the four-objective problem is provided below:
>
> | Method               | Summation       | Product         | R-IPO           | PP-DPO          | PP-IPO          |
> |----------------------|-----------------|-----------------|-----------------|-----------------|-----------------|
> | **HV ↑**            | 6279 (634)     | 6793 (659)     | 16895 (575)     | 23199 (170)     | **30861 (154)** |
> | **IGD ↓**           | 34.57 (1.35)   | 34.18 (2.07)   | 22.55 (22.55)   | 8.98 (0.55)     | **3.05 (0.44)** |
> | **Pareto Set Size ↑**| 15.67 (1.70)   | 15.3 (0.47)    | 25.5 (6.5)      | 33.5 (1.5)      | **51.5 (1.5)**  |
>
> The full results will be included in the revised paper.

---

> > ### Comment · Reviewer_UXSm · 2024-11-27
> >
> > I would like to thank the authors for their comprehensive explanations. My concerns have been well addressed, and I sincerely hope that these discussions will be included in the revised paper. So, I decided to raise my score.

---

> > > ### Author Response · Authors · 2024-11-27
> > >
> > > Dear Reviewer,
> > >
> > > Thank you for updating your rating!

---

### Author Response · Authors · 2024-11-21
**Universal rebuttal**

We apprecaiate the time and effort of all the reviewers put into reviewing our work. We have carefully considered your comments and would like to address the common concerns raised across the reviews.

---

The reviewers suggested exploring experiments with a greater number of objectives. In the current literature on multi-objective prompt optimization, benchmarks are typically limited to at most three objectives, which guided our initial choice of two or three objectives for fair comparison. However, our method is fully capable of handling problems with more objectives. To demonstrate this, we have extended our text style transfer experiment to include four objectives:

1. **Style score**
2. **Content similarity**
3. **Output fluency** (as suggested by reviewer yPkv)
4. **Conciseness** (minimizing the output length)

The results for algorithms that can be easily adapted to this four-objective setting are summarized below, based on three independent runs for each method:

| Method               | Summation       | Product         | R-IPO           | PP-DPO          | PP-IPO          |
|----------------------|-----------------|-----------------|-----------------|-----------------|-----------------|
| **HV ↑**            | 6279 (634)     | 6793 (659)     | 16895 (575)     | 23199 (170)     | **30861 (154)** |
| **IGD ↓**           | 34.57 (1.35)   | 34.18 (2.07)   | 22.55 (22.55)   | 8.98 (0.55)     | **3.05 (0.44)** |
| **Pareto Set Size ↑**| 15.67 (1.70)   | 15.3 (0.47)    | 25.5 (6.5)      | 33.5 (1.5)      | **51.5 (1.5)**  |

The full results will be included in the revised paper. These findings further demonstrate the robustness of our method in tackling multi-objective optimization problems with diverse and challenging objectives.

---

### Author Response · Authors · 2024-11-25

Dear Reviewers,

As the author-reviewer discussion period nears its deadline, we kindly ask you to review our responses to your comments, concerns, and suggestions. If you have any additional questions or feedback, we will do our best to address them promptly before the discussion period concludes. If our responses have satisfactorily addressed your concerns, we would greatly appreciate it if you could update your evaluation of our work accordingly.

Thank you once again for your valuable time and thoughtful feedback.

Sincerely,
Authors

---

### Meta-Review · Area_Chair_hdJE · 2024-12-17

**Metareview:**

This paper proposes a multi-objective prompt optimization method which makes use of a preference-based loss function inspired by DPO.

The reviewers all hold positive opinions about the paper. For example, the reviewers think that the idea of using preference-based loss is novel and inspiring, the experimental results are strong, and the paper is presented well. Some weaknesses with the original paper were missing experiments with more objectives and missing discussions with some prior related works, but these have all been addressed by the rebuttal.

All reviewers agree this is a good paper and deals with an important problem in prompt optimization, I hence recommend acceptance.

**Additional Comments On Reviewer Discussion:**

During rebuttal, the authors added experiments with more objectives which was requested by multiple reviewers, and added discussions with more related works pointed out by the reviewers. According to the reviewers' response, their concerns are all well addressed. At the end of discussion, all reviewers have positive scores for the paper.

---

### Decision · Program_Chairs · 2025-01-22

Accept (Poster)